# Mass Spectrometric Metabolic Fingerprinting of 2-Deoxy-D-Glucose (2-DG)-Induced Inhibition of Glycolysis and Comparative Analysis of Methionine Restriction versus Glucose Restriction under Perfusion Culture in the Murine L929 Model System

**DOI:** 10.3390/ijms23169220

**Published:** 2022-08-16

**Authors:** Julian Manuel Volland, Johannes Kaupp, Werner Schmitz, Anna Chiara Wünsch, Julia Balint, Marc Möllmann, Mohamed El-Mesery, Kyra Frackmann, Leslie Peter, Stefan Hartmann, Alexander Christian Kübler, Axel Seher

**Affiliations:** 1Department of Oral and Maxillofacial Plastic Surgery, University Hospital Wuerzburg, D-97070 Wuerzburg, Germany; 2Department of Biochemistry and Molecular Biology, Biocenter, D-97074 Wuerzburg, Germany; 3Fraunhofer ISC, Translational Center RT, D-97070 Wuerzburg, Germany; 4Department of Biochemistry, Faculty of Pharmacy, Mansoura University, Mansoura 35516, Egypt

**Keywords:** amino acid restriction, glucose restriction, mass spectrometry, low carb, 2-deoxy-D-glucose, 2-DG, methionine, perfusion culture, energy restriction, caloric restriction

## Abstract

All forms of restriction, from caloric to amino acid to glucose restriction, have been established in recent years as therapeutic options for various diseases, including cancer. However, usually there is no direct comparison between the different restriction forms. Additionally, many cell culture experiments take place under static conditions. In this work, we used a closed perfusion culture in murine L929 cells over a period of 7 days to compare methionine restriction (MetR) and glucose restriction (LowCarb) in the same system and analysed the metabolome by liquid chromatography mass spectrometry (LC-MS). In addition, we analysed the inhibition of glycolysis by 2-deoxy-D-glucose (2-DG) over a period of 72 h. 2-DG induced very fast a low-energy situation by a reduced glycolysis metabolite flow rate resulting in pyruvate, lactate, and ATP depletion. Under perfusion culture, both MetR and LowCarb were established on the metabolic level. Interestingly, over the period of 7 days, the metabolome of MetR and LowCarb showed more similarities than differences. This leads to the conclusion that the conditioned medium, in addition to the different restriction forms, substantially reprogramm the cells on the metabolic level.

## 1. Introduction

A cell needs two basic elements—mass and energy. The mass consists mainly of amino acids. They enable basic structures and functions in the form of proteins and enzymes and constitute the largest component of a cell’s mass [1]. For energy, the most important representatives are ATP and NAD(P)H, which are essentially obtained from carbohydrates and lipids. Glycolysis, and with it glucose, is established as one of the oldest reaction pathways for energy production, with the help of which ATP and NADH can be produced very fast [2]. For this reason, and due to its high water solubility, glucose is the central energy resource for the cells of an organism.

Mass and energy are essential for the maintenance of the cell and, to an even greater extent, for proliferation. In the case of neoplastic cells, defined by space-occupying proliferation that is almost unlimited, they, therefore, play an even greater role. For this reason, different forms of restriction are potential strategies for tumour therapy. What they all have in common is that a resource is limited to such an extent that it is no longer sufficiently available to supply the tumour cells. In the case of glucose, the approach is based on the so-called Warburg effect, which was described by Warburg and colleagues in 1924 [3]. They were able to show that tumour cells take up more glucose under aerobic conditions and metabolise it to lactate. This metabolic pathway is normally used only under an oxygen deficiency, i.e., under anaerobic conditions. At first glance, this seems inefficient, since glycolysis provides only two ATP as a net gain, while complete degradation to CO_2_ provides another 32 ATP. Although the causes of the Warburg effect are not yet understood in detail, there are a number of possibilities. For example, aerobic glycolysis enables the very rapid synthesis of ATP to maintain a high ATP level, increased synthesis of NADPH, which is increasingly needed for the biosynthesis of lipids, among other molecules, and the acidification of the tumour microenvironment, which can suppress the immune response to the tumour [4]. It is now known that the Warburg effect is generally characteristic of proliferating cells, but it plays a quantitatively and qualitatively greater role in neoplastic cells due to the nonlimited proliferation. For this reason, glucose restriction, which prevents the tumour from using glucose in any way, offers a fundamental approach to tumour therapy [5].

However, the possibilities for attacking tumours with the help of restriction have expanded considerably in the past 10 years. Although glucose is one of the central molecules in metabolism, especially in tumours, the restriction of energy and/or mass also offers a very good approach. Energy can be reduced with the help of calorie restriction, and mass can be reduced by amino acid or protein restriction [6,7,8,9]. Central to all forms of restriction is the induction of a low-energy metabolism (LEM), which is significantly mediated by the protein complex mTOR. The intracellular activity of the mTOR determines proliferation and growth. A multitude of sensors pass information about the energy and mass content to the mTOR switch point. For example, the energy equivalents ATP and NADH can be measured via the protein AMPK (5′ AMP-activated protein kinase) and with the help of sirtuins [10]. A lack of energy leads to the inhibition of the mTOR via the two proteins mentioned and results in a halt to proliferation and the induction of autophagy for the purpose of energy recycling. The content of selected amino acids is measured with the help of protein complexes; in the case of methionine, SAMTOR plays a decisive role. Here, methionine is not measured directly, but an intermediate product, S-adenosylmethionine (SAM), whose content is an indicator of the intracellular methionine concentration. However, a low methionine concentration also leads to the inhibition of the mTOR [11].

In the long term, all forms of restriction lead to astonishing results in almost all organism types, from yeast to nematodes to Drosophila to humans. In addition to the extension of the absolute lifespan, the prevention of heart and circulatory diseases, type II diabetes and cancer are prominent [7,12,13]. The restriction of glucose in the form of a reduced increase (low carb) or as absolute an avoidance as possible (ketogenic) results in similar mechanisms to calorie restriction [14].

All forms of restriction have much in common, as the reactions either use basic mechanisms of the cell or rely on evolutionarily conserved mechanisms. However, there will be equally striking differences due to the complexity. Especially with regard to the evaluation of the advantages and disadvantages of individual forms of restriction, the level of scientific knowledge is very low. This is mainly because scientific working groups usually focus only on one particular form of restriction, and thus, it is very difficult to compare the individual forms of restriction. In the case of glucose, this is further complicated by the fact that many studies do not work with a constant glucose concentration. Physiologically, in mammals, the glucose concentration in the blood is kept at a certain threshold value, which it does not fall below—in humans, approximately 3 mM, and in mice, approximately 6 mM [15]. Many experiments with glucose carried out in cell culture start with a concentration that is in the low-carb range. However, this value is not static, but continues to drop over the course of the experiment. This can lead to inconsistent results that are difficult to interpret. Experiments cannot always be carried out over a longer period of time in this way, as cells undergo apoptosis or simply die if the glucose deficiency remains below the physiological threshold [16].

The aim of our research group is to enable the comparison of individual restriction forms at the molecular level. First, we use the murine model system L929, which enables the comparison of different restrictions in one and the same system [17,18,19]. Second, we use closed, circular perfusion cultures that allow metabolite concentrations to be kept constant over longer periods of time. In this work, we used liquid chromatography mass spectrometry (LC-MS) to analyse the effect of the glycolysis inhibitor 2-deoxy-glucose (2-DG) at time points 2, 5, 8, 24, 48, and 72 h under proliferative conditions in our L929 model system. In addition, we compared methionine restriction (MetR) with glucose restriction under LowCarb conditions (3 mM) in a perfusion culture over a period of 7 days.

## 2. Results

In previous work, we already used the murine cell line L929 as a model system for the analysis of different forms of amino acid restriction [17,18,19]. The use of murine cells has several advantages. First, many studies in the field of restriction have been conducted in rodent models, and the mouse has been established as a model system for the study of energy metabolism [20]. Second, murine metabolism is up to 100× faster than human metabolism [21,22,23]. Differences in metabolism can be better analysed. Third, many of the metabolic pathways are strongly conserved in evolution. Especially in the area of restriction, many similarities can be seen from yeast to nematodes, Drosophila, rodents, primates, and humans [6,9]. mTOR and sirtuins are just two examples of these strongly conserved mechanisms [24,25].

### 2.1. Cell Division Frequency and Methionine Sensitivity of L929 Cells

Figure 1 shows the proliferation behaviour of L929 cells over a period of 120 h (Figure 1a). As described in the Materials and Methods, one can calculate the cell division frequency (f), i.e., how long a cell population takes to divide once. Figure 1b shows the values from Figure 1a, which reveal exponential growth and were used for the determination of f. For the period from 24 h to 72 h, the cell division frequency was 0.99, i.e., the cell number doubled approximately once per day. L929 thus showed very rapid growth in cell culture. The interval 72/80 was not included in the evaluation because the 80 h value no longer showed exponential growth, which can also be clearly seen in the cell division frequency of f = 0.39 for this interval (Figure 1b). The response rate to MetR was also very good (Figure 1c). Proliferation was already significantly reduced after 24 h and did not increase further during the rest of the study period.

### 2.2. Glucose Restriction and 2-DG Are Strong Inhibitors of Proliferation in L929 Cells

The main aim of this work was to establish and analyse glucose restriction in the model system L929. In the first experiment, the glucose sensitivity of the cell line and the response to the glycolysis inhibitor 2-deoxy-D-glucose (2-DG) were tested. Over a period of 96 h, log2 dilution series were generated, and absolute cell numbers were obtained every 24 h using automated digital microscopy (Pico). The results are summarised in Figure 2.

At low glucose concentrations, the proliferation of L929 cells is strongly restricted (Figure 2a). In the range of 3 mM, the proliferation rate increases strongly. The concentration of 11 mM corresponds to the standard concentration of the RPMI medium used and was, therefore, chosen as the starting concentration.

The glycolysis inhibitor 2-DG is taken up via glucose transporters and can be phosphorylated by hexokinase to 2-deoxy-glucose-6-phosphate, which cannot be further processed by glucose-6-phosphate isomerase and competitively inhibits it. In addition, hexokinase is not competitively inhibited by 2-deoxy-glucose-6-phosphate, which means that less glucose-6-phosphate is already being formed [26]. As a result, the cellular concentration of ATP decreases. In addition, 2-DG increases oxidative stress, inhibits N-linked glycosylation, and induces autophagy. It can efficiently slow cell growth and potently facilitate apoptosis in specific cancer cells. Although 2-DG itself has limited therapeutic effects in many types of cancers, it may be combined with other therapeutic agents or radiotherapy to exert a synergistic anticancer effect [27]. In the L929 cell line, 2-DG showed a clear antiproliferative effect at low concentrations (Figure 2b). Up to a concentration of 1.3 mM, the proliferation rate decreased drastically. At further increasing concentrations up to 20 mM, the cell number decreased only slightly.

### 2.3. 2-DG Works in L929 Cells at the Metabolic Level as an Energy Restriction Mimetic

In a previous work, we showed that amino acid restriction by methionine in L929 cells can induce low energy metabolism (LEM) [18]. The effect of the inhibition of glycolysis by 2-DG on the metabolome should now be analysed by liquid chromatography mass spectrometry (LC-MS). It is known that the inhibition of glycolytic processes by 2-DG is followed by a limitation of the cellular energy supply; ATP levels are reduced, and the AMP/ATP ratio is subsequently increased, leading to AMPK activation, which increases the NAD+/NADH ratio and activates SIRT1. Indeed, in rodents, adding 2-DG to the diet led to phenotypes including decreased body weight, blood glucose, insulin, body temperature, and heart rate. Moreover, 2DG was shown to induce protection against oxidative stress (overview in [28]). Last but not least, 2-DG also seems to have the potential of a caloric restriction mimetic (CRM), a substance that can induce the same effects in organisms as caloric or amino acid/protein restriction [29].

The L929 cells were analysed over a period of 72 h. Since 2-DG inhibits glycolysis very rapidly, we selected three early time points (2, 5, and 8 h) and three mid-term time points (24, 48, and 72 h). Cells were stimulated with 625 µM 2-DG. We decided on this concentration based on the results of the 2-DG experiments (Figure 2b). At this concentration, a strong antiproliferative effect can already be seen, but the inhibition is not so strong that proliferation is completely prevented, and the cells may run the risk of dying within the experimental period due to the lack of ATP. Figure 3a–c shows selected liquid chromatography mass spectrometry results summarised in groups. The complete LC-MS results, including the individual measurements, raw data, and standard deviation, are provided as the Appendix A.

As previously described, the effectiveness of the inhibitor 2-DG should be evident from the metabolites of glycolysis as well as at the level of the energy currencies. 2-DG was already effective after 2 h, and glycolysis was inhibited in the first steps (Figure 3a). Both the phosphorylated hexoses and the aldohexoses accumulated over the entire period of the experiment. In addition, the throughput rate of the subsequent metabolites up to pyruvate decreased drastically, which can be seen particularly well in the metabolites diphosphoglycerate, 3-phosphoglycerate, and phosphoenolpyruvate after 48 h and 72 h. The lactate value also decreased significantly to 51%.

The efficacy of 2-DG can also be demonstrated using different energy currencies. The AMP level was continuously high over the entire period. The ATP level collapsed after 8 h at the latest and dropped continuously to a value of approximately 1%. After 72 h, the AMP/ATP ratio, a critical parameter for the energetic state of a cell, was 1/96. In addition, the NAD(+) and NADP(+) levels remained continuously high. The levels of creatine and phosphocreatine were significantly higher than those in the control over the entire period. Phosphocreatine can be used as an additional phosphate buffer in the case of ATP deficiency, which is a classic physiological situation in the case of heavy stress within muscles.

It is noticeable that most amino acids do not show any significant difference. Exceptions are the two amino acids cysteine and glutamine (Figure 3b). Glutamine reached the maximum value of 100% after 72 h in the control and cysteine after 72 h under 2-DG.

A much clearer picture emerges with the glutamate-coupled gamma-glutamyl amino acids (Glu-Aas) (Figure 3b). These are dipeptides formed extracellularly from glutamate and another amino acid (AA), which can then be imported into the cell via different transporters. The mechanism of Glu-AA formation has become controversial. According to the classical thesis, this happens by means of the glutathione cycle. Glutathione is exported and hydrolysed, and the resulting glutamate is transferred to a free amino acid by a transferase. This produces Glu-AA and Cys-Gly. These dipeptides can then be taken up by the cell and hydrolysed to free amino acids. New glutathione is then formed from glutamate, cysteine, and glycine, and the next cycle can start [30]. However, the role of glutathione in this cycle is now critically discussed [31]. Nevertheless, Glu-AAs remain extracellularly formed dipeptides, which essentially serve to import the amino acid supply. The metabolic profile clearly shows that in the control, the two valuable resources glutamine and cysteine are imported in the form of GluGln and GluCys, whereas in the control, apart from proline (70%), all other amino acids are imported up to the maximum value of 100%.

A significant difference can also be seen in the fatty acids (Figure 3c). In the group of ethanolamines and cholines (EAs and cholines), the precursors for phospholipids and sphingomyelin, the total amount under 2-DG increases steadily after 24 h and reaches 100% for phosphocholines and glycerophosphocholines after 24 h, cholines after 48 h, and acetylcholines and phospoethanolamines after 72 h.

Basically, the analysis of the metabolome under 2-DG shows the effective and efficient blocking of glycolysis in L929 cells and the normal, more or less expected reaction of this system with regard to the inhibition of glycolysis.

### 2.4. Comparative Analysis of MetR and LowCarb Treatments of L929 in Perfusion Culture

In the next step, MetR and LowCarb were compared in the first experimental setup in the model system L929 in otherwise identical conditions. For this purpose, cells were confluently seeded in a 35 mm Petri dish, further cultured for 48 h, and then incubated in a closed perfusion system with complete medium, Met(−) (0 mg/L) or LowCarb (3 mM glucose) for 7 days in a bioreactor at 37 degrees Celsius and 5% CO_2_. Subsequently, for LC-MS analysis, 1,000,000 cells were harvested in each case, and 1 mL was taken from the respective depot media bottle of the perfusion culture. The experiment was repeated four times, and the results were summarised. Only selected results are presented below. A presentation of the entire results, including the raw data, individual measurements and the standard deviations, is attached as a Appendix A.

Perfusion culture offers great advantages. Cultivation is possible over a much longer period of time, as the nutrients are permanently supplied fresh (open perfusion) or kept constantly high (closed perfusion). Many of the results within the restriction field come from animal experiments conducted over weeks, months, and sometimes even years [7,8]. Although it is possible to analyse the early processes very well at the molecular level in simple cell culture, many mechanisms of restriction are based on longer-lasting or even permanent incubation. In addition to the time factor, however, the maintenance of the concentration of metabolites plays a decisive role. This is very well illustrated by the example of glucose. A certain glucose level is essential both in the body (blood glucose level) and within the cell culture. If the level is maintained below this threshold over a long period of time, cells undergo apoptosis [32]. In a static cell culture, the initial glucose concentration is in the low-carb range and then decreases with an increasing duration of the experiment and eventually becomes critical. In a perfusion culture, the necessary metabolite concentration can be maintained over a long period of time. In this experiment, we decided to use a closed perfusion culture. The open system offers the advantage that the cells are permanently supplied with fresh medium and the concentrations of more or less all metabolites remain constant. The closed system, on the other hand, offers the advantage that secreted metabolites accumulate and finally reach concentrations above the detection limit. In addition, communication among cells also plays a major role. Messenger substances such as growth factors and interleukins, but also simple metabolites such as lactate condition the medium and can play a decisive role. In the case of MetR, these substances include polyamines, specifically spermine or spermidine, which can be advantageous for the conversion of an amino acid restriction (AAR) [33].

One of the most striking results of comparing MetR and LowCarb in perfusion culture was that the similarities of the metabolic profile were much greater than the significant differences. Of over 170 metabolites analysed both intracellularly and extracellularly, many showed similar tendencies. This is exemplified by the essential and nonessential amino acids (Figure 4a). Most of the amino acids were at a similar level in both restriction conditions and differed only slightly even from the control, usually increasing by a few percent or between 20% and 40%. Methionine, as expected, was drastically reduced in MetR (5%), but also dropped to 73% in LowCarb. The most significant difference compared to the control was seen with cysteine, which dropped to 33% and 27%, respectively. Based on the amino acids in the supernatant, it can be seen that a large proportion of the amino acids were absorbed from the medium under each type of cultivation. The exception here was cysteine, which was secreted in large amounts. Under MetR, serine (244%), glutamine (177%), glycine (153%), alanine (151%), and threonine (150%) were also secreted to a greater extent. This was also reflected in the very high number of Glu-AAs “activated” for intracellular transport in the medium (Figure 4b). Intracellularly, the level in both restriction conditions was again quite similar, and a large proportion of the amino acids were imported more strongly than in the control. Exceptions were the GluAA of the amino acids alanine, arginine, asparagine, and, at approximately 10% less, glutamine. The lack of GluMet (0%) provided very good evidence of the efficiency of MetR. Another conspicuous feature was the acetylated cysteine (Ac-Cys) under MetR, which occurred more frequently (229%) than under the other two culture conditions.

Sulphur-based metabolism was definitely prominent in MetR. Due to the lack of a sulphur source, the entire “S” metabolism was under pressure. Within the perfusion culture, however, it was seen after 7 days that with the exception of methionine sulfoxide (19%), the cells could maintain a sulphur-based metabolism very well (Figure 4c). The situation was different under LowCarb. Here, homocysteine, a precursor of methionine or an intermediate product in the breakdown of cysteine, decreased. Taurine also decreased and hypotaurine increased; both are products of cysteine degradation. Two central metabolites are the different forms of glutathione. Both the reduced form GSH and the oxidised form GSSG decreased by approximately half compared to the control. Under MetR, GSH remained relatively constant at 109%, and GSSG increased to 154%.

Glycolysis was also regulated differently under the two forms of restriction (Figure 5a). Under MetR, the intermediates hexose-P, fructose-1,6-bisphosphate, and phosphoenolpyruvate, and the glycolysis metabolites were increased overall compared to the control; under LowCarb, hexose-P, fructose-1,6-bisphosphate, and pyruvate were significantly decreased. The two intermediates, 3-phosphoglycerate and phosphoenolpyruvate, were increased to 142% and 239%, respectively, indicating stagnation within glycolysis under LowCarb.

The differences in carbohydrates, which are involved in glycolysis processes, among other things, were also clear (Figure 5b). The metabolites UDP-glucose (UDP-Glc), UDP-glucuronate, and UDP-N-acetylglucosamine (UDP-GlcNAc) were increased between approx. 60% and 90% under MetR and reduced between approx. 30% and 45% under LowCarb. UDP-Glc is derived from the glucose-6-phosphate of glycolysis and is then modified to UDP-glucoronate. Both molecules are, therefore, fundamentally dependent on the activity of glycolysis. UDP-GlcNAc is thought to be part of the glucose sensing mechanism. There is also evidence that it plays a part in insulin sensitivity in other cells [34].

Interestingly, the TCA cycle was only marginally affected under all conditions (Figure 5c). Citrate/isocitrate was significantly higher (195%) under LowCarb than in the control and MetR. However, all other metabolites were at the same level. Under neither MetR nor LowCarb did metabolites of the citrate cycle seem to be massively used for other metabolic pathways.

Significant differences between MetR and LowCarb were also seen in the phospholipids (Figure 5d). While the phospholipid phosphoethanolamine decreased under both conditions compared to the control, the precursors of phosphatidylcholine, CDP-choline and CDP-ethanolamine, were increased under MetR by 297% and 154%, respectively, while they were strongly reduced under LowCarb by 32% and 39%, respectively. In the area of lipid metabolism, carboxylic acid acetoacetate is still conspicuous and is strongly reduced under LowCarb by 39%. Lipid metabolism seems to be fundamentally lower under LowCarb.

The polyamines in the form of spermidines can perform different functions within the cell. As already mentioned, they can be advantageous for the implementation of restriction since secreted spermidine can trigger autophagy [35]. In addition, polyamines are an indicator of the energy state of the cells. At high energy levels, large amounts of acetyl-Co are produced via glycolysis and other metabolic pathways. The degree of acetylation of, e.g., proteins, is, therefore, proportional to the energy level of the cell [36]. This is then also shown by different polyamines that are increasingly acetylated [37]. In addition, high intracellular spermine or spermidine concentrations promote protein synthesis and proliferation [38]. Basically, the different polyamines are reduced under both MetR and LowCarb and indicate a lower energy state than in the control (Figure 5e). N-acetylspermidine, at 142% under LowCarb, is an exception.

The last group presented here includes the purines, pyrimidines, and nicotinamides (Figure 5f). Here, too, many metabolites were at similar levels in all groups. In the case of purines, UMP, UDP, and CMP were significantly reduced under LowCarb conditions, while under MetR, UMP and CMP were increased by between 50% and 40%. Among purines, some metabolites were reduced under both restrictions, such as IMP and GMP. AMP was slightly increased under MetR with 110%. The nicotinamides showed the opposite trend. While mNAM and NAD(+) were increased by approximately 80–90% under MetR, both metabolites were reduced by approximately 30% under LowCarb.

### 2.5. Unlimited Cell Proliferation of L929 Cells in Closed Perfusion Culture

At the end of each experiment, the total cell numbers were determined (Figure 6). It was noticeable that in the control and under LowCarb, the cell line L929 was able to grow further and form multiple layers in the cell culture. We originally chose the experimental conditions so that the cell lawn was confluent at the beginning of the experiment and the cells were prevented from further proliferation by contact inhibition. Therefore, the analysis under this nonproliferative condition essentially examined only the effects of the different metabolite concentrations of methionine and glucose. In a previous analysis, we examined MetR under proliferative and nonproliferative conditions in normal cell culture [17]. Cell proliferation was not observed under static conditions. Instead, as shown in Figure 2c, the absolute cell number in the control even decreased slightly after 120 h. When the medium in the perfusion culture was kept at a constant high level, the L929 cells could continue to grow beyond contact inhibition under both control and LowCarb conditions. The restriction of methionine continued to severely limit growth even under perfusion conditions.

## 3. Discussion

In this work, we used liquid chromatography mass spectrometry to analyse the influence of glucose restriction using the glycolysis inhibitor 2-DG in our L929 model system to demonstrate that the system is suitable for analyses in the LowCarb condition. In a further analysis, we then compared the metabolome analyses of the two restriction conditions MetR and LowCarb in a perfusion system.

Analysis of the glycolysis inhibitor 2-DG showed a clear antiproliferative effect on L929 cells (Figure 2). In addition, as expected, the inhibitor was shown to slow glycolysis at the metabolic level downstream of the actual target, glucose-6-phosphate isomerase (Figure 3) [26]. The clearest marker for the effectiveness of 2-DG is the extreme drop in the intracellular ATP concentration to 1% (Figure 3) [27]. The conclusion is that the cell line L929 is very well suited for analysing glycolysis metabolism.

Using perfusion culture, the two restriction conditions MetR and LowCarb were compared over a period of 7 days in a closed perfusion culture. Analysis of the metabolome showed that both restriction conditions cause observable changes at the molecular level. In the case of MetR, in addition to the low methionine and GluMet contents, it is particularly noticeable that some amino acids are increasingly secreted. This is a phenomenon that we have already defined in an earlier paper as characteristic of L929 under MetR [18]. Glucose restriction was also shown to be specific at the molecular level and clearly differed from the effects induced by 2-DG, as here the concentration of the first two metabolites of glycolysis—hexose phosphate and fructose 1,6-bisphosphate—strongly decreased and then accumulated to a greater degree than 3-phosphoglycerate. The decrease in UDP-glucose, a product that, as already mentioned, is strongly dependent on the turnover rate of glycolysis, also clearly showed the changes induced by LowCarb.

Nevertheless, it is noticeable that the vast majority of the metabolites remained at similar levels, even compared to the control. However, this is also evidence of the quality of the analysis. For example, if one analyses gene expression across the transcriptome of 10,000 genes, the absolute majority of genes will not be differentially regulated. If, for example, 9000 genes were differentially regulated, this would be a strong indication of a faulty analysis. Thus, on the one hand, the many metabolites at the same level show the quality of the analysis and the stability of the perfusion culture, but on the other hand, the question arises as to what extent the different forms of restriction have been implemented consistently? In the following discussion, we want to assess to what extent the results reflect the implementation of the two restriction forms MetR and LowCarb in our system, which factors have a decisive influence on the profile, and which further analyses are useful.

The incubation period has a significant influence on the metabolic profile. It is possible that the system must be incubated for a longer period than 7 days so that the specific restriction profiles can be more extreme. On the other hand, the different forms of restriction also have many common molecular mechanisms, as already described [14]. Thus, the results can also be interpreted as a slow but sure alignment of the metabolites over time, while in the short term, the differences are much larger. Looking at the overall profile of the 2-DG analysis (Appendix A), for example, it is noticeable that at the early time points within the first 8 h, most of the metabolites are significantly different compared to those in the control.

The metabolites selected for analysis also play a role. In LC-MS analysis, we chose a setup to analyse characteristic metabolites of individual metabolic pathways, such as glycolysis or the TCA cycle. However, these approximately 170 metabolites represent only a small part of the real existing metabolites of a cell. Thus, it is possible that other metabolites not used in this analysis show a much more differentiated picture than the products we selected. Again, the argument applies that the most important metabolic pathways align over time.

The growth conditions can also influence the results. In this work, we decided to start the analyses under confluent conditions to obtain sufficient cell material for the LC-MS analyses. Furthermore, we know from previous work that the low energy status under MetR is essentially independent of proliferation or cell contact inhibition [17]. Figure 6 clearly shows that the cells continued to grow steadily within the perfusion culture. We could not observe this effect in the static cell culture. However, we assume that this effect has no negative influence on the LC-MS analyses. Again, the already mentioned argument applies that the high number of metabolites at similar levels is a quality criterion. If proliferation affected the results, the differences when comparing MetR to the control and LowCarb should be much more striking. However, the question arises as to why the cells manage to continue growing even under the LowCarb condition. Figure 2a shows the analyses performed to determine the glucose concentration used in perfusion culture. Under static conditions, the proliferation rate decreased dramatically after 96 h with an initial concentration of 3 mM glucose, but under these experimental conditions, the glucose level also continued to decrease during the experimental period. Basically, this experiment served to confirm that we can work with a level of 3 mM glucose in the perfusion culture without running the risk of cell death. After 96 h at 3 mM, the cells were observed to be alive under a microscope and proliferated significantly better than under even lower glucose concentrations. We used a medium tank in the perfusion culture that provided 60 times the volume of the contents of the Petri dish. We assumed that these conditions were suitable for L929 cells and that the cells were well supplied over this period. However, we did not want to lower the glucose concentration below 3 mM because this concentration does not correspond to the physiological conditions of the blood glucose level. In mice, 3 mM is already a very low value. Therefore, in the perfusion culture, the constant supply of fresh medium seems to be sufficient for the cells to grow steadily both in the control and under LowCarb.

The closed perfusion culture is another factor that has a strong influence on the metabolome. Basically, the closed version offers the advantage that the medium is conditioned over the experimental period. Thus, on the one hand, the medium can be included in the LC-MS analysis; on the other hand, secreted messengers may well be essential to implement the individual restriction forms. However, the exact opposite can also occur. The active substances in the medium influence the cellular events more than MetR or LowCarb. Analysis of the supernatants showed that a large number of metabolites accumulated over the experimental period. Thus, it is possible that the concentration of the secreted metabolites influences the metabolism and overlaps both MetR and LowCarb in the metabolic profile, leading to an alignment of the LC-MS profiles.

However, perfusion culture offers extreme advantages. Basically, the main advantage lies in the simulation of the blood flow of an organism, in that, on the one hand, fresh nutrients can be continuously supplied and, on the other hand, waste and possibly toxins can be removed. The differences between open and closed perfusion systems will be shown in further studies.

In principle, L929 cells are also suitable for analysing glucose metabolism. They react to 2-DG inhibition accordingly. The cells are also suitable for perfusion culture, and we were able to show that after 7 days in a closed system, the cells implemented a MetR- or LowCarb-specific metabolism. However, a large number of metabolites were at similar levels even compared to the control. For this reason, the question arises as to whether the restriction forms result in a largely similar metabolism or whether the profile is influenced more significantly by certain factors than by the restriction itself. Based on the above arguments, in future work, the cells should be analysed over a longer period of time (up to 21 days) to determine whether the metabolism continues to differentiate or remains similar. Furthermore, the experiments should be conducted in an open perfusion system to limit the effect on the restrictive factors methionine and glucose. We are sure that future analyses using the perfusion culture technique will reveal fundamental insights and mechanisms under the different forms of restriction.

## 4. Materials and Methods

### 4.1. Cell Culture

The murine fibroblast cell line L929 was purchased from the Leibniz Institute, DSMZ-German Collection of Microorganisms and Cell Cultures GmbH (Braunschweig, Germany). Cells were cultured in RPMI 1640 medium (Gibco, Life Technologies; Darmstadt, Germany) with 10% FCS (Sigma-Aldrich, Darmstadt, Germany) and 1% penicillin/streptomycin (P/S; 100 U/mL penicillin and 100 µg/mL streptomycin (Thermo Fisher Scientific, Darmstadt, Germany)) at 37 °C in a humidified atmosphere containing 5% CO_2_.

### 4.2. ImageXpress Pico Automated Cell Imaging System—Digital Microscopy (Pico Assay)

Cells were seeded at 10,000 cells in 100 µL of culture medium per well of a 96-well plate and incubated overnight. The following day, the cells were incubated in complete, methionine-free, cysteine-free, or methionine- and cysteine-free medium. The incubation time is stated in the corresponding figure legend. For staining, 10 µL of Hoechst staining solution (1:200 dilution of Hoechst 33342 (Thermo Fisher, Darmstadt, Germany) (10 mg/mL in H_2_O) in medium) was added to each well. After a 20–30-min incubation period, wells were analysed with an ImageXpress Pico automated cell imaging system (Molecular Devices, San Jose, CA, USA) via automated digital microscopy. The cells were analysed with transmitted light in the DAPI channel at 4x magnification. The complete area of every well was screened. The focus and exposure time were set via an auto setup and controlled by analysing 3–4 test wells. Finally, every result was confirmed visually, and 95% of the cells were counted and analysed.

### 4.3. Analysis of the Cell Progression Rate Using the Pico Assay

Cells were seeded at 10,000 cells in 100 µL of culture medium per well in a 96-well plate. After 24, 32, 48, 56, 72, 80, 96, 104, and 120 h, cell numbers were measured with six values for every time point, as described under Pico Assay (Section 4.2). The growth of a cell population can be described with the following formula:$$N_{t}={\;N}_{0}\times 2^{\left(t\times f\right)}$$

(N_t_ = cell number at time t; N_0_ = cell number at time 0; t = time in days (d); f = cell division frequency (1/d)).

To determine f, the formula is rearranged as follows:$$f=\frac{\left(\log\left(\frac{N_{t}}{N_{0}}\right)/\log\left(2\right)\right)}{t}$$

To obtain an overview, the measured values were first plotted as a simple diagram. From this, it was possible to see at what point the growth entered the plateau phase. Then, the values were plotted as an exponential curve, and only values in the exponential growth phase were used to determine f. In the case of L929, these were the time points 24, 32, 48, 56, and 72 h. From these values, the individual f values were calculated for the intermediate periods (Δ24/32, Δ32/48, Δ48/56, and Δ56/72). The total value f was then calculated as the mean of the four Δf values.

### 4.4. Experiments under 2-DG for LC-MS

L929 cells were seeded in 20 mL of medium in 15 cm Petri dishes and incubated overnight. A total of 2 × 10^6^ cells/Petri dish were seeded under proliferative conditions for 2, 5, and 8 h, and 1 × 10^6^ cells were seeded at 24, 48, and 72 h to prevent confluence during the test period. Every value was measured in triplicate. All media contained 10% FCS and 1% P/S [100 U/mL penicillin and 100 µg/mL streptomycin (Thermo Fisher Scientific, Darmstadt, Germany)]. After seeding, the cells were incubated with 20 mL of complete RPMI medium or 20 mL of RPMI medium with 625 µM 2-deoxy-D-glucose (2-DG) (Sigma-Aldrich, Darmstadt, Germany). Before harvesting, 1 mL of the supernatant was stored for analysis. The remaining medium was then removed, and the cells were washed with 10 mL of PBS and detached with 3 mL of trypsin/EDTA (Thermo Fisher Scientific, Darmstadt, Germany). After the addition of 7 mL of the appropriate medium, the absolute cell number in the suspensions was analysed with the automated cell counter EVE^TM^ [NanoEntek (VWR, Darmstadt, Germany)]. Each sample was measured three times, and the mean value was calculated to obtain an accurate result. Pellets with 1 × 10^6^ cells were produced by centrifugation (5 min at 1200 rpm at RT). Until the LC-MS analysis, all samples were stored at −20 °C.

### 4.5. Closed Perfusion Culture Experiments for LC-MS

For perfusion culture (Figure 7), we used RPMI 1640 (Genaxxon Bioscience, Ulm, Germany), which lacks methionine and glucose, as the basal medium. Every medium was prepared from the basal medium. All media contained 10% FCS (Sigma-Aldrich, Darmstadt, Germany) and 1% P/S [100 U/mL penicillin and 100 µg/mL streptomycin (Thermo Fisher Scientific, Darmstadt, Germany)]. For the control (complete medium), the amino acid methionine (Sigma-Aldrich, Darmstadt, Germany) was added at 15 mg/L, and glucose (Sigma-Aldrich, Darmstadt, Germany) was added at 11.1 mM. For MetR medium, glucose was added at the same concentration, and for LowCarb medium, methionine was added at 15 mg/L and glucose was added at a final concentration of 3 mM.

A total of 1 × 10^6^ cells were seeded in 35 mm dishes (Greiner Bio-One, Frickenhausen, Germany). After 48 h, cells reached confluence, the medium was removed, and the cells were stimulated with complete medium (control), MetR, or LowCarb medium. Petri dishes were incubated in a bioreactor at 37 °C in a humidified atmosphere containing 5% CO_2_. Every medium tank contained 180 mL (60-fold excess compared to the contents of the Petri dish (3 mL)). The flow rate was 0.8 mL/min, with the inflow being slower than the outflow to prevent the Petri dish from overflowing. During incubation, the Petri dish was supplied with medium using a dual perfusion set for 35 mm petri dishes (PeCon GmbH, Erbach, Germany). This consisted of a perfusion ring, which had an inlet and outlet for the medium, and a metal lid with a glass interior to prevent contamination Figure 1a–c. After 7 days, 1 mL of the supernatant was removed before harvesting and stored for analysis. The remaining medium was then removed, and the cells were washed with 10 mL of PBS and detached with 3 mL of trypsin/EDTA (Thermo Fisher Scientific, Darmstadt, Germany). After the addition of 7 mL of the appropriate medium, the absolute cell number in the suspensions was analysed with the automated cell counter EVE^TM^ [NanoEntek (VWR, Darmstadt, Germany)]. Each sample was measured four times, and the mean value was calculated to obtain an accurate result. Pellets with 1 × 10^6^ cells were produced by centrifugation (5 min at 1200 rpm at RT). Until the LC-MS analysis, all samples were stored at −20 °C.

### 4.6. LC-MS

Analysis of water-soluble metabolites in cell extracts and culture media.

#### 4.6.1. Cells

After the addition of 0.5 mL of MeOH/CH_3_CN/H_2_O (50/30/20, *v*/*v*/*v*) containing 10 µM lamivudine, cell pellets were homogenised by ultrasound treatment (10 × 1 s, 250 W output energy). Medium: One hundred microlitres of culture medium was combined with 0.4 mL of MeOH/CH_3_CN (50/30, *v*/*v*) containing 10 µM lamivudine. The external standard lamivudine was not used for absolute metabolite quantification, but was used as a quality control to compensate for eventually occurring technical issues. As quality control and for the determination of the corresponding retention times, most of the annotated metabolites (which are commercially available) were run as mixtures of pure compounds under identical experimental conditions. General procedure: The resulting suspension was centrifuged (20 kRCF for 2 min in an Eppendorf centrifuge 5424), and the supernatant was applied to a C18-SPE column that was activated with 0.5 mL of CH_3_CN and equilibrated with 0.5 mL of MeOH/CH_3_CN/H_2_O (50/30/20, *v*/*v*/*v*). The SPE eluate was evaporated in a vacuum concentrator. The resulting pellet was dissolved in 50 µL (cell extracts) or 500 µL (media extracts) of 5 mM NH_4_OAc in CH_3_CN/(25%/75%, *v*/*v*).

#### 4.6.2. LC Parameters

Mobile phase A consisted of 5 mM NH_4_OAc in CH_3_CN/H_2_O (5/95, *v*/*v*), and mobile phase B consisted of 5 mM NH_4_OAc in CH_3_CN/H_2_O (95/5, *v*/*v*).

After application of 3 µL of the sample to a ZIC-HILIC column (at 30 °C), the LC gradient programme was as follows: 100% solvent B for 2 min, a linear decrease to 40% solvent B over 16 min, maintenance at 40% solvent B for 9 min, and an increase to 100% solvent B over 1 min. The column was maintained at 100% solvent B for 5 min for column equilibration before each injection. The flow rate was maintained at 200 μL/min. The eluent was directed to the ESI source of the QE-MS from 1.85 min to 20.0 min after sample injection.

The MS parameters were as follows: scan type, full MS in positive-and-negative mode (alternating); scan range, 69–1000 *m*/*z*; resolution, 70,000; AGC-target, 3E6; maximum injection time, 200 ms; sheath gas, 30; auxiliary gas, 10; sweep gas, 3; spray voltage, 3.6 kV (positive mode) or 2.5 kV (negative mode); capillary temperature, 320 °C; S-lens RF level, 55.0; and auxiliary gas heater temperature, 120 °C. Annotation and data evaluation: peaks corresponding to the calculated monoisotopic masses (MIM +/− H^+^ ± 2 mMU) were integrated using TraceFinder software (Thermo Scientific, Bremen, Germany). Materials: Ultrapure water was obtained from a Millipore water purification system (Milli-Q Merck Millipore, Darmstadt, Germany). HPLC–MS solvents, LC–MS NH_4_OAc, and lamivudine were purchased from Merck (Darmstadt, Germany). The RP18-SPE columns were 50 mg of Strata C18-E (55 µm) in 1-mL tubes (Phenomenex, Aschaffenburg, Germany). The sonifier was a Branson Ultrasonics 250 equipped with a 13-mm sonotrode (Thermo Scientific, Bremen, Germany).

#### 4.6.3. LC-MS System

A Thermo Scientific Dionex UltiMate 3000 UHPLC system linked to a Q Exactive mass spectrometer (QE-MS) equipped with a HESI probe (Thermo Scientific, Bremen, Germany) was used. The samples were analysed with a high-resolution mass spectrometer, allowing the generation of XIC data that were analysed by applying a very narrow *m*/*z* margin (+/−3 mMU). The particle filter was a Javelin filter with an ID of 2.1 mm (Thermo Scientific, Bremen, Germany). The UPLC-precolumn was a SeQuant ZIC-HILIC column (5-μm particles, 20 × 2 mm) (Merck, Darmstadt, Germany). The UPLC column was a SeQuant ZIC-HILIC column (3.5-μm particles, 100 × 2.1 mm) (Merck, Darmstadt, Germany).

Raw data analysis and value generation (in short):

LC-MS analyses were carried out in four independent experiments at 24 h, 48 h, 72 h, 96 h, and 120 h, with each value obtained from triplicate measurements. Metabolites were quantified in cell pellets and corresponding supernatants (media) under methionine-supplemented and methionine-free conditions (12 samples per time point in total). The resulting peak areas were normalised against that of lamivudine as an external standard. From this, the mean value and standard deviation were calculated for each triplicate. For better comparisons, the values were converted to percentages. For the values of the media, the control measurement of the medium used was defined as 100%. For the cell pellets, the highest measured value in each test series within an experiment was defined as 100%. From these values, the average mean values from the four experiments were then summarised in the individual tables. For a better overview, the results were rounded to natural numbers and shown as a heatmap. The corresponding colour range is indicated individually under each table. The raw data and results for the two profiles are provided in the Appendix A.

### 4.7. Statistical Analysis

Statistical analysis was performed using GraphPad Prism 8.0 (GraphPad Software, San Diego, CA, USA). One-way ANOVA was used to compare and analyse the data of different groups, followed by the Tukey–Kramer multiple comparison test (ns; nonsignificant; * *p* < 0.05, *** *p* < 0.001).

## Figures and Tables

**Figure 1 ijms-23-09220-f001:**
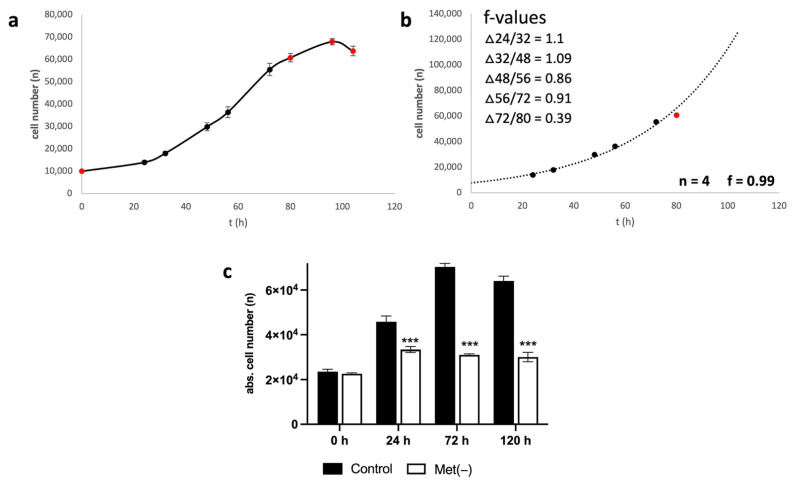
(**a**–**c**) Proliferation analysis of L929 cells. A total of 10,000 cells were seeded per well and incubated overnight. (**a**) Absolute cell numbers were analysed via ImageXpress digital microscopy as described in the Materials and Methods. The figures show a summary of the results obtained from four independent experiments (six values for every group per experiment); (**b**) trend curve of the values from figure (**a**) showing exponential growth. Values that do not represent exponential growth are shown as red dots; (**c**) L929 cells were stimulated for 24, 72, and 120 h with or without methionine. Cell proliferation was analysed via ImageXpress digital microscopy analysis as described in the Materials and Methods. The figure shows one representative experiment (five values for every group), as published previously [17]. Comparisons between the control and Met(−) groups were performed by applying one-way ANOVA followed by the Tukey–Kramer multiple comparison test (*** *p* < 0.001).

**Figure 2 ijms-23-09220-f002:**
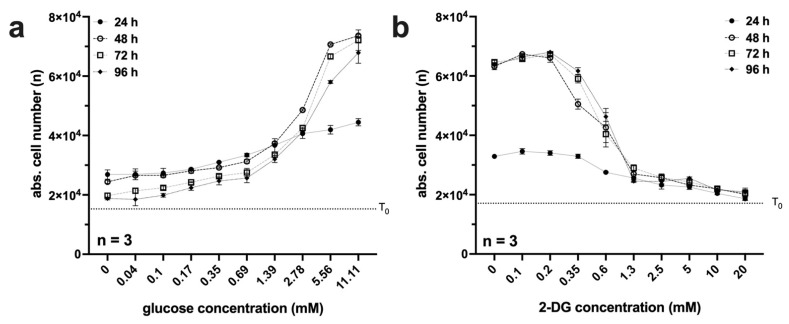
(**a**,**b**) Analysis of proliferation under glucose restriction or with the 2-DG glycolysis inhibitor in L929 cells. A total of 1 × 10^4^ cells per well were seeded on 96-well plates. On the following day, the cells were incubated in triplicate with (**a**) 11 mM glucose or (**b**) 20 mM 2-DG with a log(2)-dilution series. After 24, 48, 72, and 96 h, absolute cell numbers (n) were analysed via automated digital microscopy (Pico). The results and standard deviation of three independent experiments are shown. The cell number (T_0_) at the beginning of the experiment was also determined and is shown as a dashed line in the diagram.

**Figure 3 ijms-23-09220-f003:**
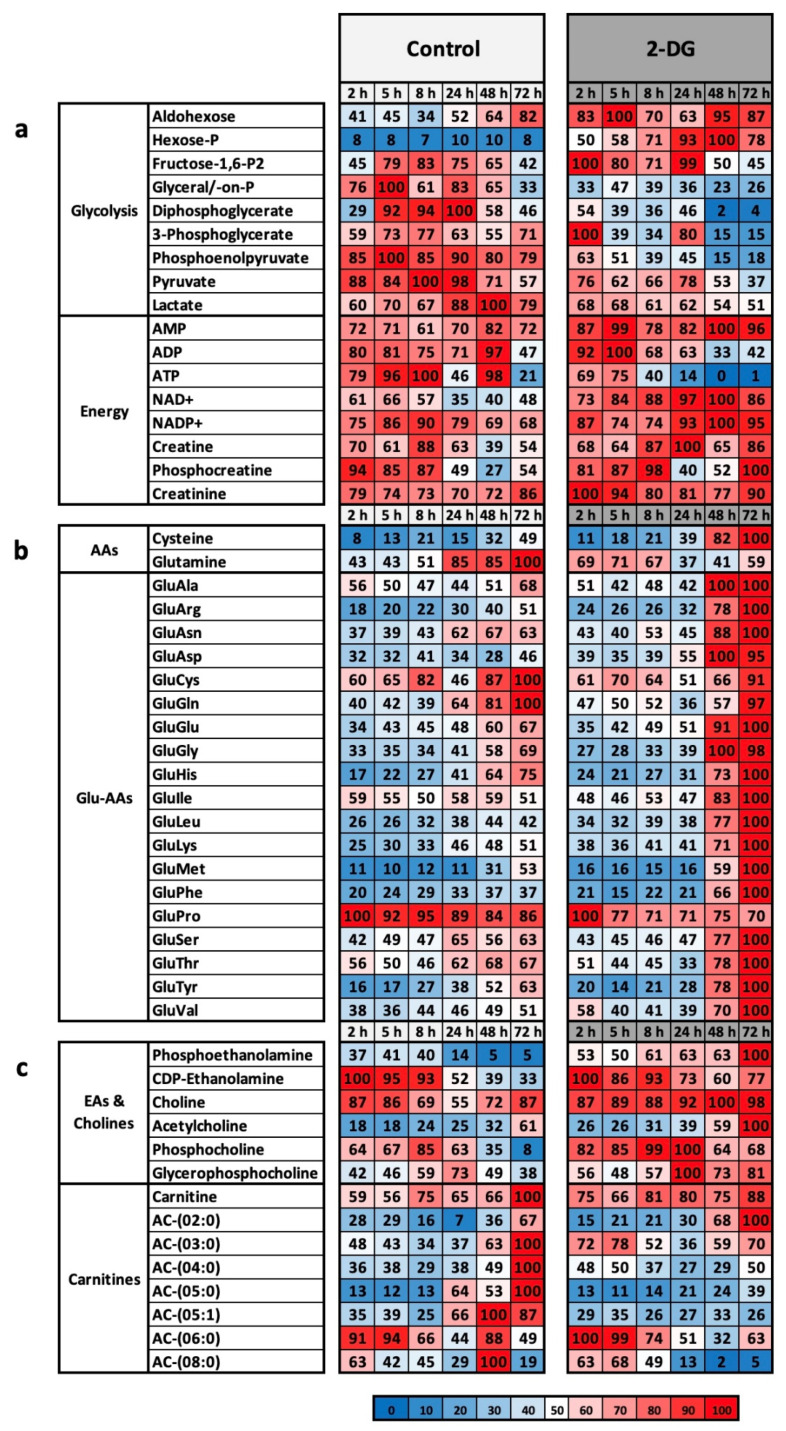
(**a**–**c**) Overview of metabolic classes and metabolic groups between control and 2-DG-stimulated cells. The metabolism of the murine cell line L929 was analysed via liquid chromatography mass spectrometry in complete medium (control) and with 625 µM 2-DG. For each time point of the experiment, the preparation was performed in triplicate. After 2 h, 5 h, 8 h, 24 h, 48 h, and 72 h, the cell lysates (intracellular) were analysed by LC-MS. The results were reproduced in three independent experiments and finally summarised. This figure shows the results of selected classes of substances and metabolic pathways for (**a**) glycolysis and energy, (**b**) amino acids (AAs), glutamine-linked amino acids (Glu-AAs), and (**c**) ethanolamines (EAs), cholines, and carnitines. For the cell pellets, the highest measured value in each test series was defined as 100%. The colour scaling is shown below the results as a legend.

**Figure 4 ijms-23-09220-f004:**
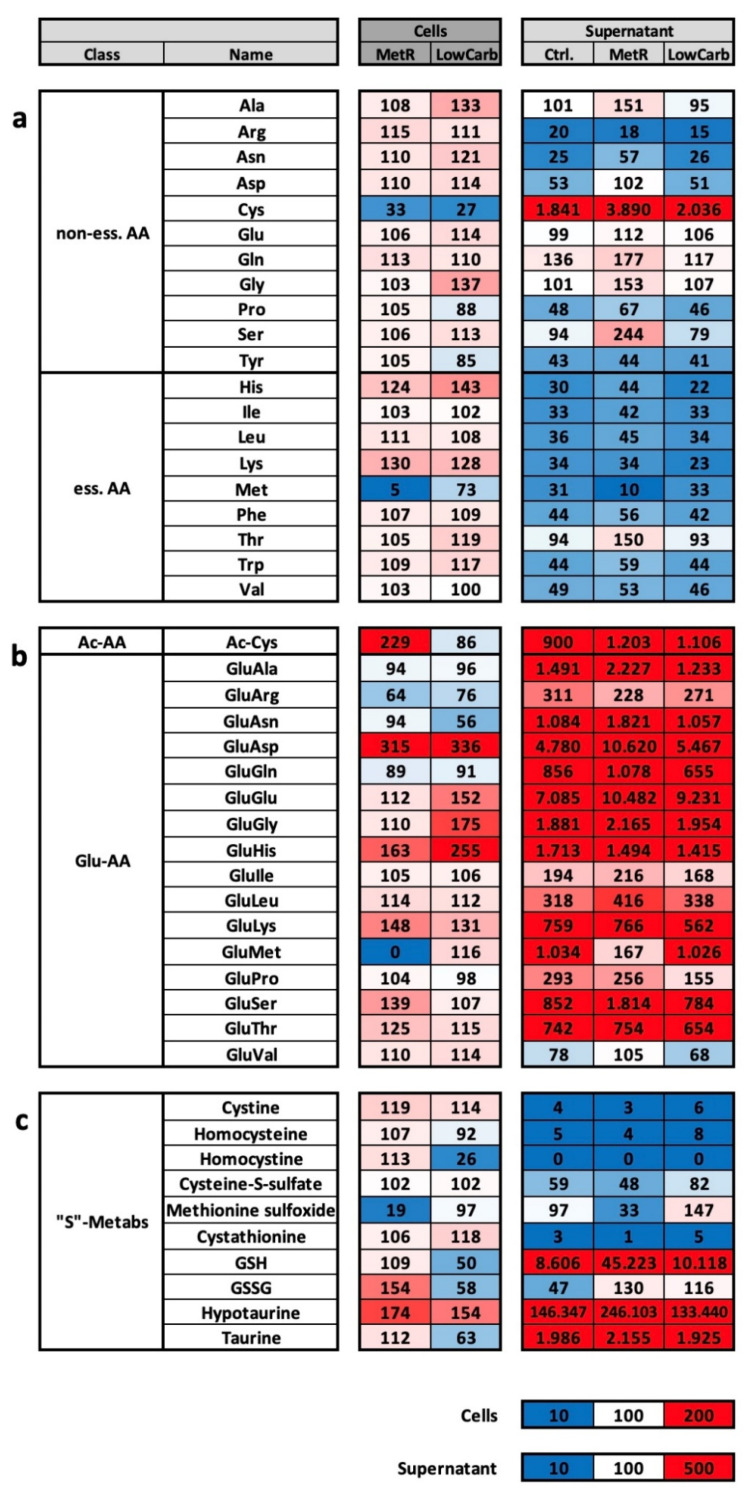
(**a**–**c**) Overview of metabolic classes and metabolic groups of liquid chromatography mass spectrometry of L929 cells in perfusion culture under MetR and LowCarb conditions. The metabolism of the murine cell line L929 was analysed via LC-MS in complete medium (control), MetR (0 mg/L), and Low Carb (3 mM glucose) for 7 days in a closed perfusion cell culture. After 7 days, the cell lysates (intracellular) and the medium (supernatant) were analysed by LC-MS. The results were reproduced in three independent experiments and finally summarised. This figure shows the results of selected classes of substances and metabolic pathways for (**a**) nonessential (non-ess. AA) and essential (ess. AA) amino acids; (**b**) acyl-linked amino acids (Ac-AA) and glutamate-linked amino acids (Glu-AA); (**c**) molecules of sulphur metabolism (“S”-Metabos). For the cell pellets, the value of the control in each test series was defined as 100%. For the values in the medium, the measurement of the control medium used was defined as 100%. The colour scaling is shown below the results as a legend.

**Figure 5 ijms-23-09220-f005:**
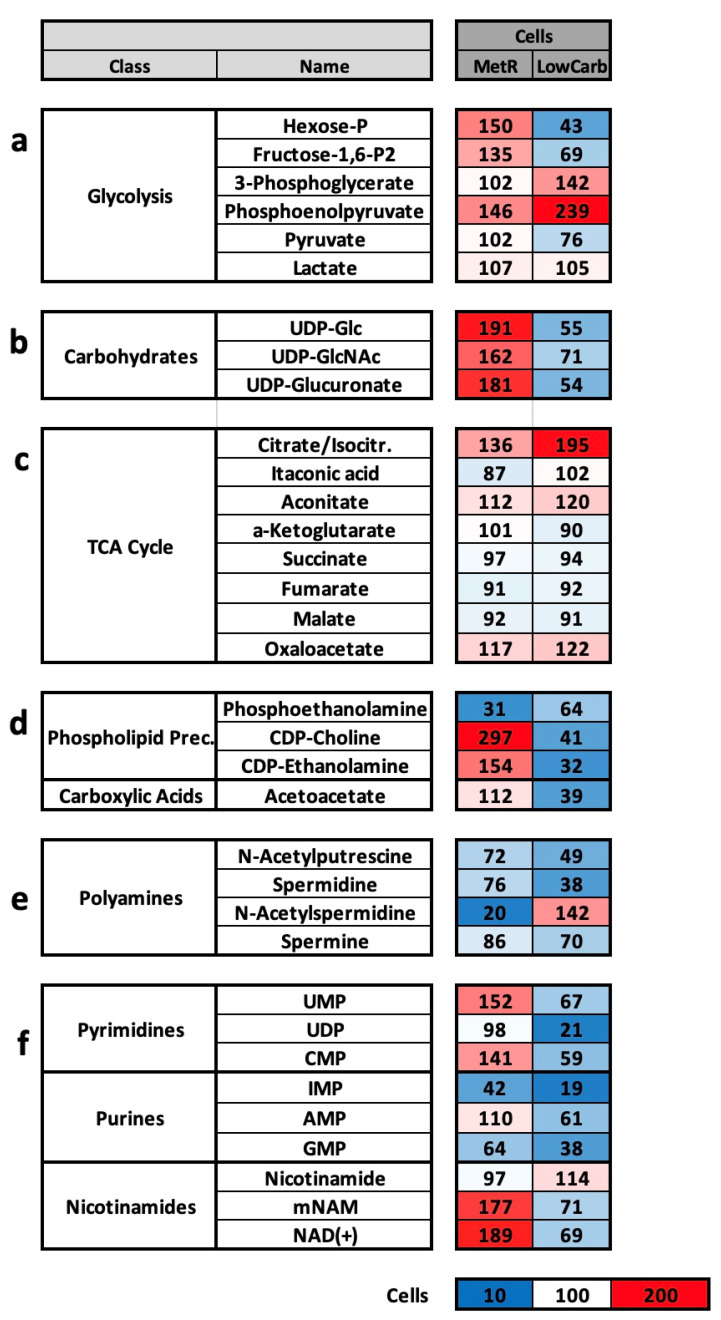
(**a**–**c**) Overview of metabolic classes and metabolic groups of liquid chromatography mass spectrometry of L929 cells in perfusion culture under MetR and LowCarb conditions. The metabolism of the murine cell line L929 was analysed via LC-MS in complete medium (control), MetR (0 mg/L), and Low Carb (3 mM glucose) for 7 days in a closed perfusion cell culture. After 7 days, the cell lysates (intracellular) and the medium (supernatant) were analysed by LC-MS. The results were reproduced in three independent experiments and finally summarised. This figure shows the results of selected classes of substances and metabolic pathways for (**a**) glycolysis; (**b**) carbohydrates; (**c**) molecules of the tricarboxylic acid (TCA) cycle; (**d**) phospholipid precursors; (**e**) polyamines; (**f**) pyrimidines, purines, and nicotinamides. The value of the control in each test series was defined as 100%. The colour scaling is shown below the results as a legend.

**Figure 6 ijms-23-09220-f006:**
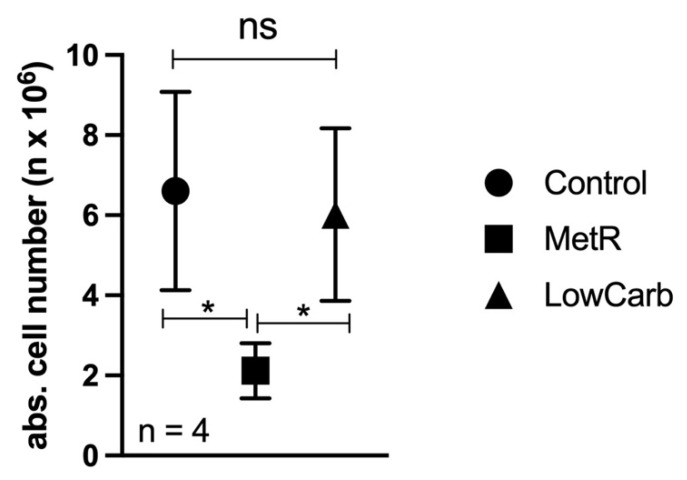
Final cell number in perfusion culture under different conditions. Towards the end of the experiment, the absolute cell number of each Petri dish was determined using an EVE^TM^ automated cell counter. The mean values of the four experiments and the respective range of cell counts are plotted here. Comparisons between the control and MetR groups were performed by applying one-way ANOVA followed by the Tukey–Kramer multiple comparison test (* *p* < 0.05, ns = not significant).

**Figure 7 ijms-23-09220-f007:**
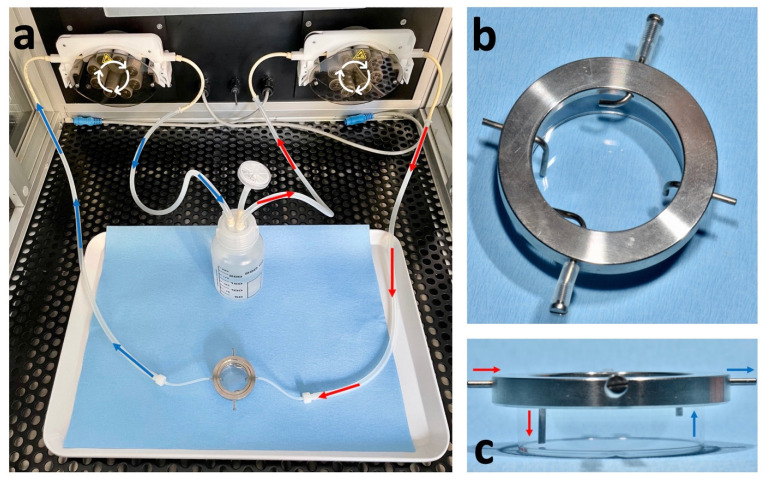
(**a**–**c**) Closed perfusion culture and dual perfusion set for 35 mm petri dishes. (**a**) Examples for a closed perfusion for one Petri dish. The inside of the bioreactor is shown. The red line shows the inlet, and the blue line shows the outlet. The tubes run through separate pump systems. This allows the outlet to be faster than the inlet and thus prevents the petri dish from overflowing; (**b**) top view; (**c**) side view of the dual perfusion set for 35 mm petri dishes. The system has a total of 4 inlets and outlets. In our case, only two were used, and the other two were closed. The red arrows show the influx, and the blue arrows show the outflux, including the height of the medium in the petri dish.

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
