# Peer review of "Mass Spectrometric Metabolic Fingerprinting of 2-Deoxy-D-Glucose (2-DG)-Induced Inhibition of Glycolysis and Comparative Analysis of Methionine Restriction versus Glucose Restriction under Perfusion Culture in the Murine L929 Model System"

_ijms, 2022, doi:10.3390/ijms23169220_

Round 1

Reviewer 1 Report

The manuscript entitled Mass Spectrometric Metabolic Fingerprinting of 2-Deoxy-D-Glucose (2-DG)-Induced Inhibition of Glycolysis and Comparative Analysis of Methionine Restriction versus Glucose Restriction under Perfusion Culture in the Murine L929 Model System reports the comparison of individual restriction forms at the molecular level. For this purpose, liquid chromatography mass spectrometry (LC-MS) was used to analyze the effect of the glycolysis inhibitor 2-deoxy-glucose (2-DG) at different times under proliferative conditions in L929 model system. This manuscript should be accepted in International Journal of Molecular Sciences after minor revisions.

The abstract should be rewritten since should contain aim, methodology and main results. Moreover, “mass spectrometry (LC/MS)” should be liquid chromatography mass spectrometry (LC-MS). Line 33: which means LM/CS??

Author Response

First of all, we would like to thank the reviewers for reading and evaluating the paper and for the positive feedback.

We have rewritten the abstract accordingly and used the term liquid chromatography mass spectrometry and the abbreviation LC-MS throughout the text.

Thank you very much for your support,

Yours sincerely,
Axel Seher

Reviewer 2 Report

The manuscript entitled “Mass Spectrometric Metabolic Fingerprinting of 2-Deoxy-D- 2 Glucose (2-DG)-Induced Inhibition of Glycolysis and Comparative Analysis of Methionine Restriction versus Glucose Restriction under Perfusion Culture in the Murine L929 Model System” by Volland demonstrated the perfusion culture to keep the concentration of metabolites as constant as possible over a period of 7 days in order to compare methionine restriction (MetR) and glucose restriction (LowCarb) in the same system. Then they analyzed the metabolome in L929 cells by mass spectrometry (LC/MS) and also analysed the inhibition of glycolysis by 2-deoxy-D-glucose over a period of 72 h using LM/CS. This is an interesting study and written well.

I have a few minor comments.

Figures and tables fond should be increased for a better view to readers.

The advantages of this model should be discussed with comparable studies. 

Author Response

First of all, we would like to thank the reviewers for reading and evaluating the paper and for the positive feedback.

We have edited the figures and tables fonds and enlarged them accordingly for a better view. 

We have added a paragraph in the discussion that again shows the advantages of our model. However, we have not compared it with published studies.

We believe that one runs the risk of comparing apples and oranges, as the differences in the individual studies can be considerable. Starting with the cells used, media, time periods, etc. If you have noticed any studies that would make a good comparison, please feel free to provide us with the citations and we will include them in the discussion.

Thank you very much for your support,

Yours sincerely,
Axel Seher